# Hybrid Multilevel Thresholding Image Segmentation Approach for Brain MRI

**DOI:** 10.3390/diagnostics13050925

**Published:** 2023-03-01

**Authors:** Suvita Rani Sharma, Samah Alshathri, Birmohan Singh, Manpreet Kaur, Reham R. Mostafa, Walid El-Shafai

**Affiliations:** 1Department of Computer Science and Engineering, Sant Longowal Institute of Technology and Engineering, Longowal, Sangrur 148106, Punjab, India; 2Department of Information Technology, College of Computer and Information Sciences, Princess Nourah Bint Abdulrahman University, P.O. Box 84428, Riyadh 11671, Saudi Arabia; 3Department of Electrical and Instrumentation Engineering, Sant Longowal Institute of Technology and Engineering, Longowal, Sangrur 148106, Punjab, India; 4Department of Information Systems, Faculty of Computers and Information, Mansoura University, Mansoura 35511, Egypt; 5Security Engineering Lab, Computer Science Department, Prince Sultan University, Riyadh 11586, Saudi Arabia; 6Department of Electronics and Electrical Communications Engineering, Faculty of Electronic Engineering, Menoufia University, Menouf 32952, Egypt

**Keywords:** brain tumor, multilevel thresholding, optimization algorithm, segmentation

## Abstract

A brain tumor is an abnormal growth of tissues inside the skull that can interfere with the normal functioning of the neurological system and the body, and it is responsible for the deaths of many individuals every year. Magnetic Resonance Imaging (MRI) techniques are widely used for detection of brain cancers. Segmentation of brain MRI is a foundational process with numerous clinical applications in neurology, including quantitative analysis, operational planning, and functional imaging. The segmentation process classifies the pixel values of the image into different groups based on the intensity levels of the pixels and a selected threshold value. The quality of the medical image segmentation extensively depends on the method which selects the threshold values of the image for the segmentation process. The traditional multilevel thresholding methods are computationally expensive since these methods thoroughly search for the best threshold values to maximize the accuracy of the segmentation process. Metaheuristic optimization algorithms are widely used for solving such problems. However, these algorithms suffer from the problem of local optima stagnation and slow convergence speed. In this work, the original Bald Eagle Search (BES) algorithm problems are resolved in the proposed Dynamic Opposite Bald Eagle Search (DOBES) algorithm by employing Dynamic Opposition Learning (DOL) at the initial, as well as exploitation, phases. Using the DOBES algorithm, a hybrid multilevel thresholding image segmentation approach has been developed for MRI image segmentation. The hybrid approach is divided into two phases. In the first phase, the proposed DOBES optimization algorithm is used for the multilevel thresholding. After the selection of the thresholds for the image segmentation, the morphological operations have been utilized in the second phase to remove the unwanted area present in the segmented image. The performance efficiency of the proposed DOBES based multilevel thresholding algorithm with respect to BES has been verified using the five benchmark images. The proposed DOBES based multilevel thresholding algorithm attains higher Peak Signal-to-Noise ratio (PSNR) and Structured Similarity Index Measure (SSIM) value in comparison to the BES algorithm for the benchmark images. Additionally, the proposed hybrid multilevel thresholding segmentation approach has been compared with the existing segmentation algorithms to validate its significance. The results show that the proposed algorithm performs better for tumor segmentation in MRI images as the SSIM value attained using the proposed hybrid segmentation approach is nearer to 1 when compared with ground truth images.

## 1. Introduction

Brain tumors, cancerous or noncancerous, are an outgrowth of abnormal cells in the brain. Malignant brain tumors are possible but rare [1]. Malignant brain tumors have a non-uniform structure and contain active (cancer) cells, while benign brain tumors have a uniform structure and are not cancerous [2]. Tumor identification, treatment planning, and monitoring of response to ontological therapy for brain tumors rely heavily on the reliable quantification and morphology of tumors derived from imaging data [3]. Magnetic Resonance Imaging (MRI) is a noninvasive medical imaging technique [4]. MRI produces high-quality images of human organs in 2D and 3D formats. Owing to its high-resolution images on brain tissues, the MR imaging modality is regarded to be one of the most accurate techniques for MRI categorization [5] and is also used to identify many disorders due to its image quality. MRI is the most common method to examine brain tissues that have been infected. Different operations are applied to MRI images for the detection of brain tumors. Brain tumor segmentation is one important method which plays a crucial role in the detection of brain tumors. Gray matter (GM), white matter (WM), and cerebrospinal fluid (CSF) are the normal brain tissues that are separated from the tumor tissues (active tumor, edema, and necrosis) during brain tumor segmentation [6]. Brain tumors are notoriously difficult to segment due to their various appearances in terms of location, size, forms, and recurrence [7,8,9].

Different researchers have proposed several methods for brain tumor segmentation. Joseph and Singh used k means and morphological operation methods for the segmentation of the MRI image [10]. Rehman et al. proposed a segmentation approach for the MRI images based on deep autoencoder-decoder [11]. Toufiq et al. utilized an optimized threshold difference algorithm and rough set methods for the segmentation task [12]. Tripathi et al. proposed an automatic segmentation method based on deep learning, cross-channel normalization, and parametric rectified linear units for the segmentation of the brain tumor [13]. Bodapati et al. proposed a segmentation approach using two channel based deep learning model [14]. Maqsood et al. proposed a multi-model system for segmentation using deep learning and a multi-class support vector machine [15].

The deep learning-based models are widely used and are quite successful [16]. However, the deep learning methods extensively depend on the size of the dataset and model performance degrades when there is a data distribution difference between the training data and test data. These models have parameter setting problems due to the presence of a large number of trainable parameters. Additionally, data collection at a high level is often time consuming, expensive, or even not possible in different scenarios. Thus, in such cases these models become inefficient. Apart from the deep learning-based segmentation methods, threshold-based approaches are used for the segmentation task by selecting the optimal threshold values. The threshold-based approaches are characterized by their ease of implementation and ability to give accurate segmentation results [17]. It can be divided into two types: bi-level and multi-level threshold. In the bi-level category, a single threshold value is used in the prior category to separate the image into two homogeneous foreground and background areas. While in the multi-level category, these are utilized to segment an image into more than two areas based on pixel intensities, known as histogram [18]. When segmenting an image, determining the thresholding values is very important due to the presence of enormous image thresholds; hence, this topic demands more investigation. This motivates us to propose a novel method for selecting the optimal multilevel thresholding values for segmentation of the brain tumors and performance enhancement for brain tumor detection.

The remaining paper is organized as follows: Section 2 provides the related work. The proposed approach has been described in Section 3. In Section 4, the dataset used in this work has been detailed. The experimental results and discussions are given in Section 5. The conclusions and future scope of the work are given in Section 6.

## 2. Related Work

In this section, segmentation methods based on the multilevel threshold values have been analysed. Two approaches are commonly used to determine optimal threshold values for segmenting a given image into several regions. These are Otsu method [19] and Kapur entropy [20]. Otsu method maximizes the between-class variance, while Kapur entropy maximizes the entropy of the classes. These techniques are applicable for determining a single threshold value. However, it is impossible to precisely determine ideal threshold values for multi-level in these approaches. Consequently, multi-level thresholding is considered a challenge that needs to be optimized. The relevant literature makes extensive use of meta-heuristic approaches to solve these challenges.

Given their versatility and ease of implementation, academics have extensively demonstrated metaheuristic algorithms’ ability to handle complex real-world issues, such as tracking of objects [21], feature weighting [22], feature selection [23,24], improvement of machine learning algorithms [25], monitoring [16], and engineering optimization algorithms [26,27], etc., in recent decades. Metaheuristic algorithms, in contrast to deterministic approaches, do not rely on gradient information to discover optimal solutions in the search space, instead, the randomly generated search agents and specialized operators. Many natural phenomena served as inspiration for these operators. Consequently, there are primarily three types of metaheuristic algorithms: (1) swarm-based, (2) natural evolution-based, and (3) physics-based methods. The two main approaches of study for multilevel image segmentation are the classical approach and the meta-heuristic approach. An incredible amount of progress has been made in the field of image segmentation during the past few decades. Traditional approaches to multilevel image thresholding have been proven to be inefficient due to the lengthy time required to find the optimal values with which to maximize the objective function. As a result, the computational time issue of multilevel thresholding algorithms for image segmentation is successfully addressed by a number of evolutionary metaheuristic algorithms in the literature.

Oliva et al. proposed a multilevel thesholding method for the segmentation of the digital images based on the harmony search optimization algorithm [28]. Oliva et al. utilized an electromagnetism-like algorithm for the selection of optimal threshold values of the images [29]. Kandhway and Bhandari introduced energy curve and the minimum cross entropy and multiverse optimizer algorithm-based multilevel threshold selection approach [30]. Upadhyay and Chhabra proposed Kapur’s entropy and crow search optimization algorithm-based multilevel thresholding method [31]. Rather and Bala proposed constriction coefficient-based particle swarm optimization and gravitational search algorithm to find the optimal threshold values by utilizing the strength of both algorithms [32]. Resma and Nair proposed kill herd optimization for the segmentation of the images by maximizing the values of Kapur and Otsu entropy [33]. Houssein et al. utilized black widow optimization and the best threshold configuration using Otsu or Kapur as an objective function for the optimal threshold selection of the images [34].

Existing multilevel thresholding methods are only available for generalized images. The existing multilevel thresholding methods based on optimization algorithms are not utilized for the segmentation of the medical images because of their variability and complexity. Additionally, the optimization algorithm has the problem of slow convergence speed and can stuck in local optima. Considering these as motivation, in this work, a novel Dynamic Opposite Bald Eagle Search (DOBES) optimization algorithm has been proposed which is an improved version of the Bald Eagle Search (BES) [35] algorithm. The modifications are applied to solve the problem of slow convergence speed and local optima stagnation of BES algorithm. Using this DOBES algorithm a hybrid Multilevel Thresholding Image Segmentation method has been proposed to find the optimal threshold values and the additional undesired regions of the segmented image are further removed using the morphological operations based post-processing procedure.

The main contributions of this work are as follows:The DOBES algorithm is proposed by invoking the DOL method in the initialization, as well as exploitation, phases of the BES algorithm to solve the problems of slow convergence speed and local optima stagnation.A hybrid multilevel threshing approach is proposed for the segmentation of the brain tumor.The proposed hybrid segmentation approach is compared with state-of-the-art algorithms to show its significance.

## 3. Proposed Hybrid Multilevel Thresholding Image Segmentation Approach

In this paper, a hybrid multilevel threshold segmentation approach has been proposed for the detection of brain tumors in the MRI image. The hybrid approach has two phases: Proposed Dynamic Opposite Bald Eagle Search (DOBES) optimization based multilevel threshold selection, and morphological operations-based post-processing procedure. The generalized block diagram of the proposed approach has been depicted in Figure 1.

The hybrid approach has utilized the proposed DOBES algorithm in the first phase to find the optimal threshold levels. The selected optimal threshold levels are utilized to generate the threshold image based on the different threshold levels. The binary segmented image of the threshold image is generated for the next phase of operations. In the second phase, the morphological operations are applied to the previously generated binary segmented image to find out the area of interest and neglect the other undesired regions. The details of the different phases of the proposed approach are as follows.

### 3.1. DOBES Based Multilevel Threshold Selection

Image thresholding is classified into two types: bilevel and multilevel. The multilevel technique is the progression from the bilevel approach [36]. For a bimodal gray-level histogram with one valley between two peaks, the bi-level thresholding process is computationally simple and straightforward. The multilevel thresholding approach, on the other hand, is significantly more computationally demanding, but it might be well suited to a multimodal gray-level histogram with several peaks and troughs [37]. However, as the number of necessary thresholds rises, multilevel thresholding becomes more complex, and it becomes considerably more difficult when working with a two-dimensional gray-level histogram. To overcome this issue, metaheuristic optimization algorithms have been developed, which yield exceptionally better results for different types of images.

In this paper, a Dynamic Opposite Bald Eagle Search (DOBES) optimization algorithm is used for the selection of the optimal multilevel threshold of the brain MRI image. This algorithm is an improved version of the Bald Eagle Search (BES) [35] optimization algorithm. The BES algorithm has problems of slow convergence and local optima stagnation. These problems are addressed in the DOBES algorithm by employing Dynamic Opposition Learning (DOL). The BES algorithm has three phases: Select, Search, and Swoop. The select phase is used for exploring the available whole search space to search for the solution. Whereas the search phase is used to exploit the selected area, and the swoop phase is used to target the best solution. The formulation of the three phases is as follows:(1)$$\begin{array}{c} P_{i,new}=P_{best}+\alpha \times r\left(P_{mean}-P_{i}\right) \end{array}$$

In this equation, *i* is the total number of search agents, random variable *r* have value ranging from $\left[0,1\right]$, and α is between $\left[1.5,2\right]$.
(2)$$\begin{array}{c} P_{i,new}=P_{i}+m\left(i\right)\times \left(P_{i}-P_{i+1}\right)+l\left(i\right)\times \left(P_{i}-P_{mean}\right)\\ where,\;\;l\left(i\right)=\frac{lr(i)}{max(\left|lr\right|)}\;\;and\;\;m\left(i\right)=\frac{mr(i)}{max(\left|mr\right|)}\\ lr(i)=r(i)\times \sin(\theta (i)),\;\;mr(i)=r(i)\times \cos(\theta (i))\\ \theta (i)=\alpha \times \pi \times rand,\;\;r(i)=\theta (i)+R\times rand \end{array}$$
where, α is a algorithmic parameter having values in between 5 and 10, rand have values in between 0 and 1, and *R* denotes the number of search cycles having value in between 0.5 and 2.
(3)$$\begin{array}{c} P_{i,new}=rand\times P_{best}+l1\left(i\right)\times \left(P_{i}-c1\times P_{mean}\right)\\ +m1\left(i\right)\times \left(P_{i}-c2\times P_{best}\right)\\ l1\left(i\right)=\frac{lr(i)}{max(\left|lr\right|)}\;\;and\;\;m1\left(i\right)=\frac{mr(i)}{max(\left|mr\right|)}\\ lr\left(i\right)=r\left(i\right)\times \sinh\left[\theta (i)\right],\;\;mr\left(i\right)=r\left(i\right)\times \cosh\left[\theta (i)\right]\\ \theta (i)=\alpha \times \pi \times rand,\;\;r(i)=\theta (i) \end{array}$$
where c1,c2 are the algorithmic numbers having values in the range of $\left[1,2\right]$.

The proposed DOBES approach employs DOL to improve the initialization of the search agents. The initialization technique influences both the rate of convergence and the time necessary to find the best solution. As a result, DOL has been implemented in the DOBES algorithm to improve the likelihood of convergence to the global optimum while avoiding the stagnation problem associated with local optima. Furthermore, DOL has been used to accelerate the convergence rate by more equally spreading the search phases. DOL is utilized during the exploitation phase (the search phase) to analyse both the candidate solutions and their corresponding opposing candidate solutions, extending the exploitation space and improving the chance of discovering a better solution. The flow chart of the DOBES algorithm is given in Figure 2.

The working of the DOBES algorithm (Figure 2) starts with the parameter setting of the algorithm. Then, in the modified initialization phase randomly initial positions of the search agents have been generated, and using the DOL method opposite positions of the search agents have been determined. Fitness values have been calculated for the search agents and only the best search agents are selected. In the select phase, the positions of the search agents are updated to explore the search space and selection of the best area. The selected area has been extensively searched in the modified search phase. In the modified search phase, the DOL positions of the search agents are considered to improve the working of the exploitation phase. In the last phase, the best optimal solutions have been selected. These phases are repeated until a stopping criterion has been satisfied.

#### Fitness Function for Multilevel Thresholding

For the multilevel thresholding Kapur’s entropy [20] which is based on the probability distribution of the image’s histogram is used as a fitness function in the proposed DOBES algorithm. The formulation of Kapur’s method is as follows:(4)$$\begin{array}{c} F_{kap}=\sum_{i=0}^{k}H_{i},H_{i}=-\sum_{j=t_{i}}^{j=t_{i+1}}\frac{P_{j}}{A_{j}}ln\left(\frac{P_{j}}{A_{j}}\right) \end{array}$$
where $P_{j}$ is the probability of the gray-levels.

The DOBES algorithm has utilized Kapur’s entropy as a fitness function to find the multilevel threshold levels in the MRI image.

### 3.2. Morphology-Based Post-Processing Procedure

The morphological operations are used in the proposed hybrid approach to remove the additional undesired areas which are available in the binary segmented image. The operations used in this work are for the edge detection, image dilation, boundary detection, unwanted area removal, and area filling. Figure 3 shows the block diagram of the post-processing method adapted in this work.

In Figure 3, the binary segmented image of the previous phase is used as an input image. The Canny edge detection method [38] has been utilized in the first step to detect all the edges of segments, then the image dilation method [39] has been used to fill the gaps by adding pixels in the edges of the selected area. In the next step, boundary detection method [40] has been applied to fetch the boundaries of the regions and the small undesired regions are removed from the binary segmented image. The image filling morphological operation [41] has been used to fill the remaining enclosed area regions. In the end, the region of tumor is selected from the processed image.

The brain image processing layout of the proposed approach has been shown in Figure 4.

The layout shows that the optimal multilevel threshold values have been first computed using the proposed DOBES optimization algorithm and using the selected multilevel threshold values a threshold image is generated. Further, a binary segmented image has been generated using the threshold image to detect the tumor region. After the generation of the binary segmented image, a post-processing procedure based on the morphological operations has been applied to select the tumor region and remove other undesired regions.

## 4. Dataset Description

The brain tumor dataset has been taken from the Figshare website having URL [https://figshare.com/articles/dataset/brain_tumor_dataset/1512427 (accessed on 15 November 2022)]. The dataset contains T1-weighted images of 233 patients. The images contain three types of tumor which are meningioma, glioma, and pituitary tumor. Figure 5 shows the input and ground truth images of the brain MRI.

In Figure 5, different types of MRI image views, i.e., Axial, Sagittal, and Coronal have been depicted. In the figure, three types of tumors and their respective ground truth image representation of tumor locations in the brain.

## 5. Results and Discussions

A hybrid segmentation approach for the segmentation of tumor regions from the MRI image has been proposed in this paper. The segmentation tests are performed on a computer equipped with the Windows 10 Pro operating system, an Intel^®^ Xenon^®^ CPU E5-2650 v3 (2.30 GHz), 8 GB of RAM, and MATLAB 2019a platform.

The performance metrics used for the analysis of the proposed segmentation model are structured similarity index measure (SSIM), peak signal-to-noise ratio (PSNR), fitness value, mean square error (MSE), and standard deviation. Mean square error (MSE) quantifies the error at each pixel position and generates a mean value, which is then used to calculate the image’s PSNR. MSE is the difference in intensity between the input image and the segmented image. Peak signal-to-noise ratio (PSNR) is defined as the ratio of the highest achievable signal power to introduced noise. It is used to assess the quality of an image’s reconstruction. The SSIM is a perception-based approach that takes image deterioration into account as a perceived change in structural information. SSIM is commonly used to determine the relationships between input and segmented images.

### 5.1. Comparison of Proposed DOBES and BES Algorithm for Benchmark Images

The performance of the proposed DOBES algorithm has been analysed and compared with the original BES algorithm to show the significance of the proposed DOBES algorithm. For the performance comparison five benchmark images (Baboon, Boat, Cameraman, Couple, Male) have been selected. The images have been taken from the UCS-SIPI image dataset having URL [https://sipi.usc.edu/database/ (accessed on 27 January 2023)]. The values of the performance measures are provided in Table 1. These results are obtained after running the algorithm for 100 iterations and 10 reruns with a population size of 30.

Table 1 shows a performance comparison of the proposed DOBES algorithm and BES algorithm for the benchmark images. The DOBES algorithm attains higher PSNR values for the baboon, boat, cameraman, and couple images while the BES algorithm attains higher PSNR values only for the male image. The DOBES algorithm attains SSIM metric values close to 1 and higher than the BES algorithm which shows that the DOBES algorithm performs better than the BES algorithm. This proves that the proposed DOBES algorithm is significantly better.

### 5.2. Analysis of Proposed Hybrid Segmentation Approach for the Brain Images

The proposed hybrid segmentation approach has been tested on the Figshare MRI image database for the tumor area detected from the MRI images. In Figure 6, the input, threshold image, binary segmented image, final segmented image after morphological operations, and segmented images mapped with the ground truth images have been depicted for the BES and proposed hybrid segmentation approach. The morphological operations-based post-processing approach has been applied to the segmented images generated using BES optimization algorithm for a fair comparison of these algorithms.

Figure 6 represents the segmentation performance of the BES and the proposed hybrid segmentation approach. From the figure, it has been observed that the BES algorithm fails to identify the optimal threshold values of the MRI image, and the tumor region is not identified. This can be observed in the image generated after mapping with the ground truth image. The pink part in the image shows the original ground truth value of the tumor region and the green part in the image shows the tumor region selected using the BES algorithm. In comparison to the BES algorithm, the proposed hybrid segmentation approach is able to successfully select the optimal threshold values, and the tumor region is segmented correctly. This is proved using the ground truth mapping image.

The threshold levels and the convergence curves obtained using the proposed segmentation approach have been shown in Figure 7.

Figure 7 depicts three threshold values in the histogram image of the MRI in the (a) part of the figure. From (a), it has been observed that the threshold values are optimal as the changes in the intensity level of the pixel values are correctly identified. In (b), the convergence curve has been plotted to show the convergence speed of the proposed hybrid segmentation approach.

For the performance comparison of the proposed approach, three existing multilevel thresholding optimization algorithms are selected, i.e., Constriction Coefficient Based Particle Swarm Optimization and Gravitational Search Algorithm (CPSOGSA) [32], Electromagnetism-like Algorithm (EMO) [29], Harmony Search (HS) optimization [28]. Additionally, the proposed algorithm is compared with the BES algorithm to show the significance of the proposed DOBES algorithm.

In Figure 8, multilevel thresholding images and the corresponding binary images have been shown for the optimization algorithms.

From Figure 8, it has been observed that the DOBES algorithm has segmented the tumor region clearly from the other brain regions whereas the BES and the state-of-the-art optimization algorithms fail to find the optimal threshold values for the tumor segmentation which leads to merging of the tumor region with the brain regions. This proves that the proposed algorithm is better in comparison to the comparative optimization algorithms.

In Table 2, a comparison of the proposed hybrid approach with the existing methods for the brain tumor detection has been shown. The same morphological operations-based post-processing procedure has been applied to all the state-of-the-art algorithms for a fair comparison of the segmented brain tumor images.

From the table, it has been observed that the proposed algorithm attains better values for the metrics ’best fitness values’, as well as ’mean fitness values’, in comparison to the BES and the other existing algorithms. Additionally, the proposed segmentation approach attains the highest or comparable SSIM values in the case of the segmented images vs. ground truth images.

Thus, the multilevel threshold values which have been obtained using the proposed approach are optimal as detection of the tumor regions in the brain MRI images is close to ground truth values. Additionally, the visual analysis proves that the existing algorithms are not able to find the optimal threshold values especially in the cases where the contrast difference between foreground and the background regions are not quite distinctive.

## 6. Conclusions and Future Directions

Brain tumors, a leading cause of death worldwide, are abnormal growths of tissue inside the skull that can impede the nervous system and body function. In neurology, segmenting brain MRIs is a foundational first step with multiple uses, including quantitative analysis, operational planning, and functional imaging. In this work, we apply the hybrid segmentation approach having two phases for the task of segmenting brain tumors from the MRI images. The selection of optimal multilevel threshold values has been accomplished using the Dynamic Opposite Bald Eagle Search (DOBES) optimization algorithm in the first phase and morphological operations-based post-processing procedure is utilized in the second phase for the selection of tumor regions. The proposed DOBES algorithm is a development by improving the original Bald Eagle Search (BES) algorithm. The original BES method has the issues of slow convergence and local optima stagnation. These problems are resolved in the DOBES algorithm by employing Dynamic Opposition Learning (DOL). Then the morphological operations based post-processing procedure has been applied for the tumor segmentation of the MRI image. The performance of proposed DOBES algorithm has been analysed using the benchmark images. The proposed DOBES algorithm achieves better PSNR and SSIM values in comparison to the BES algorithm for the benchmark images. The proposed hybrid segmentation approach is compared with the existing algorithm for the performance analysis. The SSIM values attained using the proposed hybrid segmentation approach for the segmented vs. ground truth images prove that the segmented images obtained using the proposed hybrid approach are closer to the ground truth images. Thus, the results show that the proposed approach successfully finds the optimal threshold values for the segmentation of the tumors.

In the future, the proposed method can be used to find the optimal thresholds in RGB images. Additionally, in place of Kapur’s method other variance schemes can be used as objective functions. Medical data analysis is now a very active area of research and a fertile application domain for machine learning. As a result, the proposed approach can be used to find optimal multilevel thresholding values from more complex medical images and feature selection to improve the classification performance.

## Figures and Tables

**Figure 1 diagnostics-13-00925-f001:**
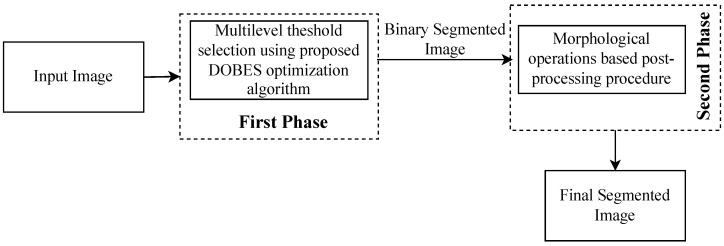
Generalized layout of proposed hybrid segmentation approach.

**Figure 2 diagnostics-13-00925-f002:**
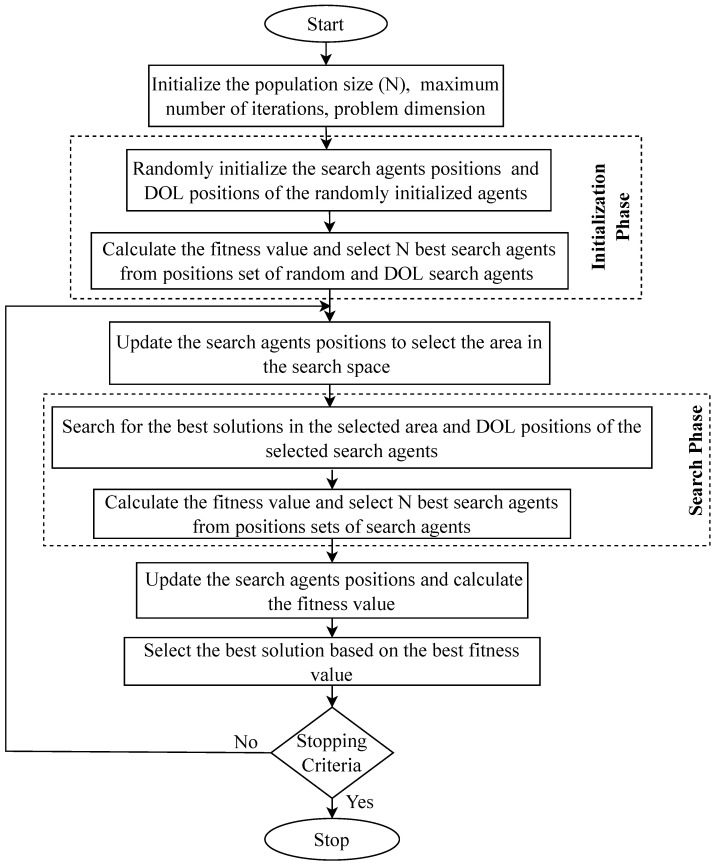
DOBES algorithm.

**Figure 3 diagnostics-13-00925-f003:**
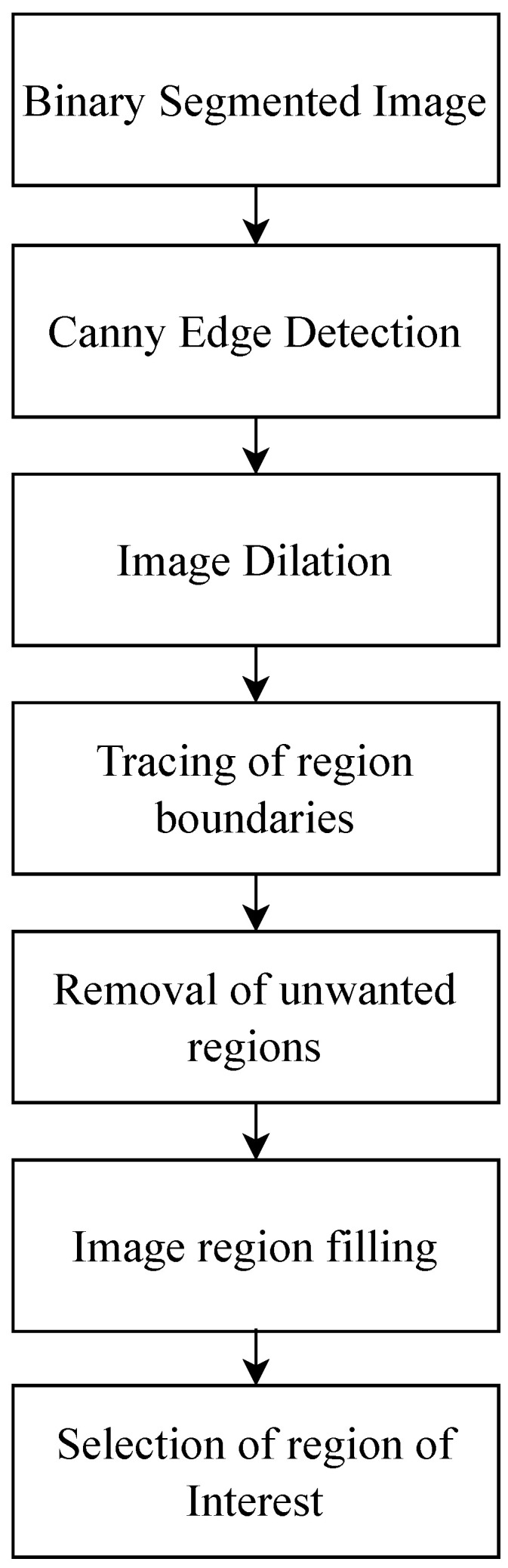
Block diagram of morphological based post-processing.

**Figure 4 diagnostics-13-00925-f004:**
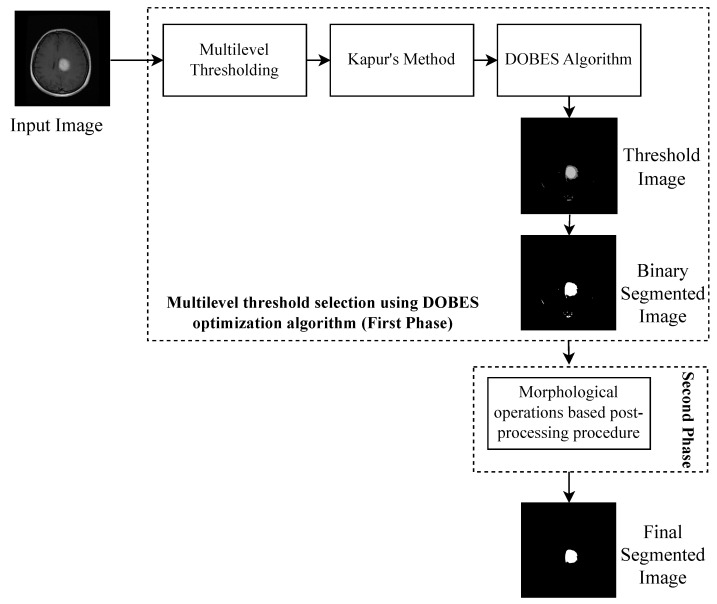
Image processing layout of proposed hybrid multilevel thresholding image segmentation approach.

**Figure 5 diagnostics-13-00925-f005:**
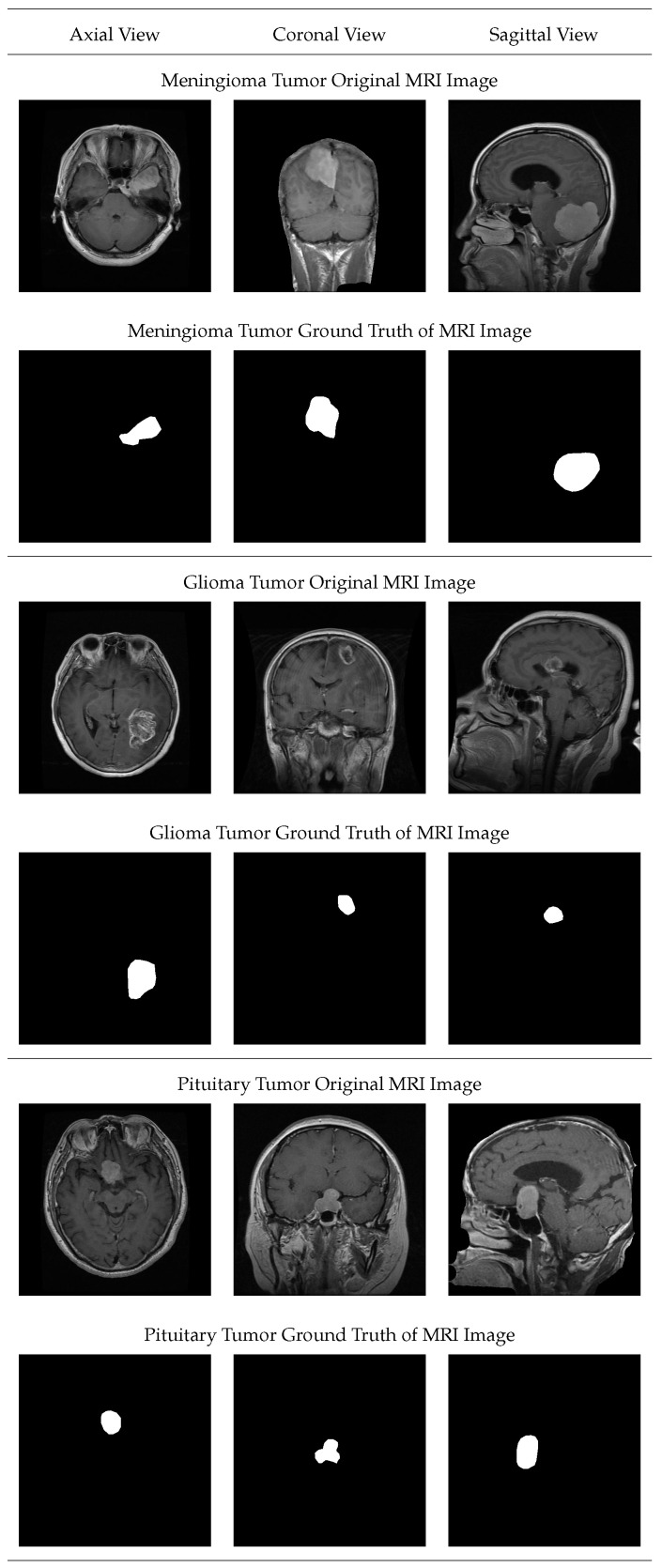
Visual representation of the brain MRI and respective ground truth images.

**Figure 6 diagnostics-13-00925-f006:**
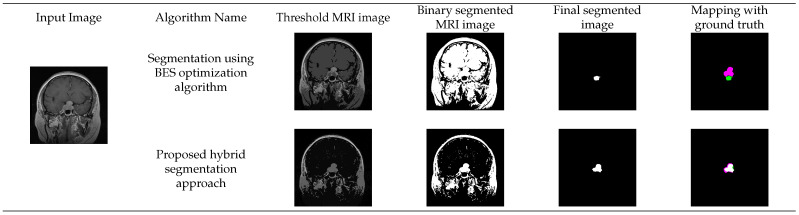
Visual representation of the MRI images generated at different phases of proposed hybrid segmentation approach.

**Figure 7 diagnostics-13-00925-f007:**
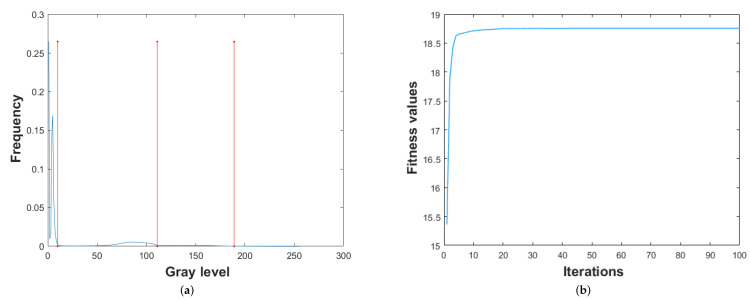
Multilevel threshold and convergence curve of proposed hybrid segmentation approach. (**a**) Multilevel thresholds and (**b**) Convergence curve.

**Figure 8 diagnostics-13-00925-f008:**
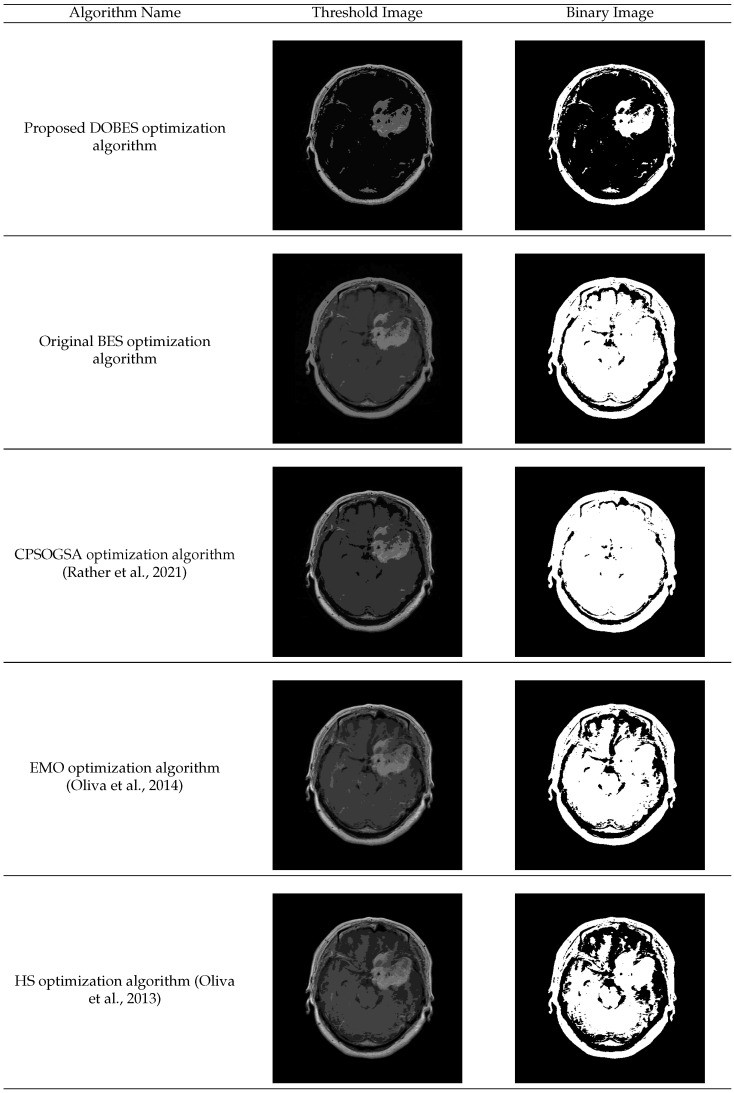
Threshold and binary image of the optimization algorithms obtained using optimization algorithms. DOBES = Dynamic Opposite Bald Eagle Search optimization algorithm, BES = Bald Eagle Search optimization algorithm, CPSOGSA = Constriction Coefficient Based Particle Swarm Optimization and Gravitational Search Algorithm, EMO = Electromagnetism-like Algorithm, HS = Harmony Search optimization [28,29,32].

**Table 1 diagnostics-13-00925-t001:** Evaluation metrics of the proposed DOBES and original BES algorithm for benchmark images.

Image Name	Algo Name	Optimal Level	Best Fitness Value	Mean Fitness	Standard Deviation	MSE	PSNR	SSIM
Baboon	DOBES	[1, 38, 76, 79, 115, 143, 160, 255]	44.5825	41.2860	1.8289	3.90 × $10^{3}$	22.2251	0.9028
	BES	[1, 61, 69, 115, 127, 176, 185, 240]	43.2949	41.0181	2.4316	1.05 × $10^{3}$	17.9447	0.7972
Boat	DOBES	[1, 50, 91, 107, 128, 176, 181, 255]	45.0538	43.4139	1.4986	5.86 × $10^{3}$	20.4532	0.7886
	BES	[1, 1, 78, 107, 109, 142, 229, 253]	44.5601	42.7437	1.6598	7.06 × $10^{3}$	19.6419	0.7593
Cameraman	DOBES	[18, 59, 100, 128, 146, 193, 197, 254]	44.6614	42.7581	0.9297	4.05 × $10^{3}$	22.0586	0.7587
	BES	[19, 45, 94, 97, 146, 147, 197, 254]	43.9419	41.5759	2.0517	5.56 × $10^{3}$	20.6824	0.7063
Couple	DOBES	[2, 37, 67, 76, 119, 124, 170, 255]	43.9526	41.4636	1.1683	2.90 × $10^{3}$	23.5099	0.5521
	BES	[2, 3, 67, 77, 77, 119, 180, 189]	41.8404	40.1056	1.7395	7.47 × $10^{3}$	19.3993	0.4066
Male	DOBES	[2, 37, 67, 76, 119, 124, 170, 255]	45.6260	43.7534	1.4264	6.15 × $10^{3}$	20.2439	0.7806
	BES	[1, 56, 66, 108, 123, 172, 174, 250]	43.6013	43.4959	1.6096	5.62 × $10^{3}$	20.6373	0.7598

**Table 2 diagnostics-13-00925-t002:** Evaluation metrics of the proposed hybrid segmentation approach.

Tumor Name	Image Number	Algo Name	Optimal Level	Best Fitness Value	Mean Fitness	Standard Deviation	MSE	PSNR	SSIM (Input Image vs. Threshold Image)	SSIM (Segmented Image vs. Ground Truth)
Meningioma	1	Proposed hybrid segmentation approach	[7,32,97,98,150,159,208,209,255]	47.9206	45.4620	1.7862	5.02 × $10^{2}$	21.1280	0.6750	0.9998
		BES	[7,8,97,99,120,172,172,244,245]	47.4714	44.8024	2.3320	1.07 × $10^{3}$	17.8513	0.5552	0.9998
		CPSOGSA [32]	[8,30,98,129,135,172,202,212,246]	44.5615	42.1077	1.7931	5.12 × $10^{2}$	21.0401	0.6665	0.9998
		EMO [29]	[9,33,57,83,111,142,172,202,225]	44.3197	44.2593	0.0754	8.67 × $10^{1}$	28.7517	0.7650	0.9931
		HS [28]	[8,14,59,89,103,135,158,199,217]	42.1911	41.8324	0.2809	1.51 × $10^{2}$	26.3402	0.7319	0.9927
	177	Proposed hybrid segmentation approach	[7,7,110,110,110,188,188,253,255]	46.0736	44.4955	1.5804	1.35 × $10^{3}$	16.8425	0.4624	0.9999
		BES	[7,9,11,110,112,112,182,222,229]	45.3021	43.2352	2.5435	1.20 × $10^{3}$	17.3250	0.4791	0.9975
		CPSOGSA [32]	[38,58,69,107,108,129,135,168,201]	43.5696	41.1617	1.4121	1.09 × $10^{2}$	27.7433	0.6206	0.9973
		EMO [29]	[10,38,63,87,110,138,166,192,220]	44.1964	44.1614	0.0422	6.47 × $10^{1}$	30.0209	0.6547	0.9974
		HS [28]	[14,31,87,105,114,143,170,190,227]	42.0029	41.4002	0.5052	2.38 × $10^{2}$	24.3740	0.5914	0.9967
	660	Proposed hybrid segmentation approach	[7,8,83,83,130,144,145,255,255]	48.8510	47.7180	1.5629	1.33 × $10^{3}$	16.8945	0.4046	0.9997
		BES	[8,8,83,114,132,144,178,245,252]	48.8194	46.1231	1.7590	1.28 × $10^{3}$	17.0730	0.3872	0.0.9888
		CPSOGSA [32]	[8,8,110,114,117,170,194,206,225]	47.9582	42.7699	2.7136	1.92 × $10^{3}$	15.2996	0.2862	0.9951
		EMO [29]	[16,52,83,110,133,156,179,202,226]	44.7735	44.6824	0.0878	1.94 × $10^{2}$	25.2540	0.6314	0.9913
		HS [28]	[18,39,49,74,87,145,157,188,204]	41.9134	41.9134	0.0000	1.50 × $10^{2}$	26.3837	0.6498	0.9916
Giloma	719	Proposed hybrid segmentation approach	[8,88,89,132,142,173,198,213,253]	48.3412	45.8864	1.8913	1.00 × $10^{3}$	18.1200	0.5247	0.9999
		BES	[8,8,90,94,115,161,167,252,254]	47.8164	44.8459	2.3497	1.04 × $10^{3}$	17.9696	0.5205	0.9999
		CPSOGSA [32]	[8,36,60,104,108,118,148,165,218]	44.5352	40.5329	2.5836	1.29 × $10^{2}$	27.0192	0.7309	0.9959
		EMO [29]	[10,38,63,89,115,142,168,195,222]	44.5907	44.5582	0.0728	9.33 × $10^{1}$	28.4307	0.7228	0.9958
		HS [28]	[26,41,53,87,100,149,186,208,229]	42.5611	42.5611	0.3693	1.35 × $10^{2}$	26.8168	0.7018	0.9961
	799	Proposed hybrid segmentation approach	[10,11,94,95,140,156,159,255,255]	48.2201	46.6109	2.7537	1.33 × $10^{3}$	16.8896	0.5149	0.9999
		BES	[4,11,94,94,124,156,157,227,255]	48.1490	44.7736	2.2807	1.31 × $10^{3}$	16.9455	0.5821	0.9969
		CPSOGSA [32]	[11,41,92,109,121,141,162,171,225]	43.4365	40.8422	1.9584	3.70 × $10^{2}$	22.4465	0.6895	0.9999
		EMO [29]	[17,43,67,92,117,142,166,191,217]	44.0809	44.0121	0.1339	1.15 × $10^{2}$	27.5277	0.7183	0.9989
		HS [28]	[11,26,60,84,89,122,150,180,216]	41.8422	41.8422	0.0000	1.40 × $10^{2}$	26.6661	0.7505	0.9986
	895	Proposed hybrid segmentation approach	[6,6,81,108,111,141,193,255,255]	47.9738	46.3416	1.9144	1.42 × $10^{3}$	16.6180	0.3920	0.9999
		BES	[6,7,80,111,112,130,193,206,237]	47.8793	45.2731	2.2741	1.35 × $10^{3}$	16.8191	0.4138	0.9977
		CPSOGSA [32]	[6,6,66,109,115,117,154,170,236]	46.3880	42.3097	2.5580	7.08 × $10^{2}$	19.6282	0.5084	0.9990
		EMO [29]	[27,53,79,103,125,149,172,193,213]	44.7794	44.6589	0.0832	1.71 × $10^{2}$	25.8036	0.6951	0.9985
		HS [28]	[17,35,63,91,138,148,164,182,219]	42.9729	42.5060	0.0000	1.60 × $10^{2}$	26.1007	0.7308	0.9986
Pituitary	59	Proposed hybrid segmentation approach	[8,8,91,92,145,168,168,255,255]	49.2296	46.8252	2.6490	1.82 × $10^{3}$	15.5255	0.3652	0.9997
		BES	[8,90,91,92,157,157,168,238,245]	48.1580	45.2139	2.0382	1.78 × $10^{3}$	15.6157	0.3687	0.9981
		CPSOGSA [32]	[9,88,91,122,137,154,159,212,248]	45.7014	43.1113	2.3225	1.62 × $10^{3}$	16.0319	0.3885	0.9982
		EMO [29]	[14,40,65,91,119,146,173,200,226]	45.2712	45.1999	0.1596	1.54 × $10^{2}$	26.2647	0.7128	0.9982
		HS [28]	[26,48,58,70,94,110,124,183,227]	41.9849	41.9453	0.1430	1.26 × $10^{2}$	27.1156	0.7223	0.9983
	1391	Proposed hybrid segmentation approach	[7,7,79,80,109,131,132,248,255]	45.5724	44.3092	1.1470	1.02 × $10^{3}$	18.0536	0.5370	1.0000
		BES	[7,15,79,84,108,126,132,192,214]	45.3682	43.9759	1.4224	7.36 × $10^{2}$	19.4634	0.6091	0.9986
		CPSOGSA [32]	[7,20,78,88,116,132,139,146,200]	43.5107	40.8545	1.6167	5.72 × $10^{2}$	20.5536	0.6519	0.9999
		EMO [29]	[10,33,55,78,101,124,147,170,193]	42.0512	41.6049	0.2809	8.78 × $10^{1}$	28.6962	0.7966	0.9980
		HS [28]	[11,43,76,97,137,146,161,167,183]	39.2840	38.4971	0.9387	1.84 × $10^{2}$	25.4932	0.7513	0.9982
	1405	Proposed hybrid segmentation approach	[8,8,97,97,106,182,188,255,255]	46.2615	44.8582	1.1550	1.48 × $10^{3}$	16.4229	0.5128	0.9998
		BES	[8,8,97,106,121,189,198,254,255]	46.0916	42.9182	2.3297	1.44 × $10^{3}$	16.5362	0.5179	0.9998
		CPSOGSA [32]	[9,38,90,93,93,129,159,163,184]	44.3048	41.7757	1.9603	4.08 × $10^{2}$	22.0293	0.7199	0.9995
		EMO [29]	[11,38,64,91,115,140,165,188,212]	43.6460	43.5694	0.1596	1.14 × $10^{2}$	27.5647	0.8004	0.9987
		HS [28]	[25,67,83,91,131,155,173,179,213]	40.0705	40.0705	0.0000	1.82 × $10^{2}$	25.5385	0.7476	0.9985

DOBES = Dynamic Opposite Bald Eagle Search optimization algorithm, BES = Bald Eagle Search optimization algorithm, CPSOGSA = Constriction Coefficient Based Particle Swarm Optimization and Gravitational Search Algorithm, EMO = Electromagnetism-like Algorithm, HS = Harmony Search optimization.

## Data Availability

The data presented in this study are openly available in Figshare website at URL [https://figshare.com/articles/dataset/brain_tumor_dataset/1512427] and USC-SIPI image database at URL [https://sipi.usc.edu/database/].

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
