# Peer review of "Hybrid Multilevel Thresholding Image Segmentation Approach for Brain MRI"

_diagnostics, 2023, doi:10.3390/diagnostics13050925_

Round 1

Reviewer 1 Report

·         In the literature survey section, the problems of the existing methods are not clearly explained and the results of the work related to them are not properly explained.

·         Explanation of  figures are not clear.

·         In the abstract, the authors should place more results and values on this part. Add more explanation in conclusion part.

·         Give more explanation in existing method about result and discussion part.

·         The abstract also should be revised according to the main idea of this research and the main motivation behind the proposed research.

·         The research findings and contribution need to be stated clearly. As well as the obtained results in this paper.

·         The related works section is very short, and no benefits from it. I suggest increasing the number of studies and adding a new discussion to show the studied works' advantages, disadvantages, and weaknesses. The authors should discuss the literature review more deeply and clearly.

·         Add a new figure to show the general procedures of the proposed method.

·         Check all the figures and others to make sure you call the figure before it appears.

·         It is not a clear result of the forecasting method between the proposed method and previous work. I hope the authors should show the result.

·         Abbreviations are confusing, and some of them did not well defined.

·         The authors need to use more passive voice

·         Miss use of definite and indefinite articles

·         The paper has to be proofread by a native English reviewer.

·         Authors should also include more scientific logic in explaining the experimental results.

·         In general, update the references lists by the following reference related to:

·         After that, you should mention that these methods can be used to optimize the problem.

·       The author needs to include a benchmarking table in the performance section.

·       The manuscript must re-check typographical and reference citation errors and correct them.

cite the below

1.      Raju, K. Lova, S. Koteswara Rao, Rudra Pratap Das, M. Nalini Santhosh, and A. Sampath Dakshina Murthy. "Passive target tracking using unscented kalman filter based on monte carlo simulation." Indian Journal of Science and Technology 8, no. 29 (2015): 1-7. 10.17485/ijst/2015/v8i29/76981

  Murthy, A. Sampath Dakshina, T. Pavani, and K. Lakshmi. "An Application of Firefly Hybrid Extended Kalman Filter Tracking a Reentry Object." Indian Journal of Science and Technology 9 (2016): 28. 10.17485/ijst/2016/v9i28/84214

Author Response

Thank you for your valuable suggestions. All the comments are properly addressed.

Reviewer 2 Report

This work utilizes a new algorithm for the segmentation of brain MRI images, with the aim of detecting different types of cancer. Results indicate that the new algorithm (DOBES) is superior to other methods used in the past.

I request the authors to include the matlab script of the algorithm in a free data repository or in supplementary data in the journal if possible.  This will help other researchers to use the algorithm for additional, potentially lifesaving work.  

Author Response

Thank you for the valuable comments. All the comments are properly addressed.

Round 2

Reviewer 1 Report

·         The authors should elaborate more on their methodology's novelty and superiority, which should be clearly articulated in the response letter.

·         Check the paper language and make sure all language errors have been fixed.

·         Please use the complete form of abbreviations in the abstract.

·         The abstract also should be revised according to the main idea of this research and the main motivation behind the proposed research.

·         The research findings and contribution need to be stated clearly. As well as the obtained results in this paper.

·         The related works section is very short, and no benefits from it. I suggest increasing the number of studies and adding a new discussion to show the studied works' advantages, disadvantages, and weaknesses. The authors should discuss the literature review more deeply and clearly.

·         Add a new figure to show the general procedures of the proposed method.

·         Check all the figures and others to make sure you call the figure before it appears.

·         It is not a clear result of the forecasting method between the proposed method and previous work. I hope the authors should show the result.

·         Abbreviations are confusing, and some of them did not well defined.

·         The authors need to use more passive voice

·         Miss use of definite and indefinite articles

·         The paper has to be proofread by a native English reviewer.

·         Authors should also include more scientific logic in explaining the experimental results.

·         In general, update the references lists by the following reference related to:

·         After that, you should mention that these methods can be used to optimize the problem.

·         What other possible methodologies can be used to achieve your objective in relation to this work?

·         Overall, the organization of the paper is good and the authors have given their efforts in working on this domain and problems. Yet, this work could be more strengthened by incorporating the suggestions and the paper might be clear and readable for researchers who work in interdisciplinary domains.

 cite below papers 

Sampath Dakshina Murthy, Achanta, Thangavel Karthikeyan, and R. Vinoth Kanna. "Gait-based person fall prediction using deep learning approach." Soft Computing (2021): 1-9. https://doi.org/10.1007/s00500-021-06125-1.

Achanta, Sampath Dakshina Murthy, Thangavel Karthikeyan, and R. Vinoth Kanna. "Wearable sensor based acoustic gait analysis using phase transition-based optimization algorithm on IoT." International Journal of Speech Technology (2021): 1-11.https://doi.org/10.1007/s10772-021-09893-1

Author Response

Thank you for the comments. All comments are properly addressed and the paper is updated accordingly.

Reviewer 2 Report

Well done

Author Response

Thank you for the comments and suggestions.

Round 3

Reviewer 1 Report

Over all it is good